# Nitrogen and phosphorus fertilisers optimise root morphology and soil nutrients in mixed annual grass and bean sown grassland in alpine regions

**Ting-Xu Feng[1], Fei Li[1], Xue-Mei Xiang[1], Wei-Shan Lin[1], Xi-Jie Wei[2], Lin Zhang[2], Ke-Jia De [1,2]***

**1** Qinghai University, Xining City, Qinghai Province, China, **2** Qinghai Academy of Animal Husbandry and Veterinary Science, Qinghai-Xining, China

\* dekejia1002@163.com

**Data availability statement:** All relevant data are within the manuscript and its Supporting Information files.

**Funding:** This study was financially supported by Chengduo County's Research and Demonstration of Annual Forage Seed Propagation and Silage Processing and Storage Technology in the form of a grant

## Abstract

The grass-bean hybrid system in alpine regions was an important method to increase the root system and nutrients of grasses, however, there was no clear conclusion from the research on whether the nitrogen——phosphorus fertiliser additions in alpine regions had an enhancement effect on the root system and nutrients of the grass-bean hybrids. Through the establishment of an artificial grass-bean mix in an alpine region at 4,270 meters above sea level, we examined the effects of exogenous fertilizer additions on forage roots and nutrients in an annual grass-legume forage mix system using (fertilizer application: $A_{1\sim4}$ vs. mix ratio $B_{1\sim5}$).The results illustrate that: the addition of nitrogen and phosphorus did not significantly increase forage nutrients in grass-bean mixtures compared to no fertiliser, which appeared to be more favourable; nitrogen——phosphorus nutrients increased the root length and root projection area of forage, and also increased the nutrient content of the soil.The forage CP, TN, TC, and TP contents as well as the soil nutrient contents gradually increased as the fraction of legume forage increased under various mixing ratios. However, as the percentage of legumes increased, root morphology trended downward, and mixed treatments outperformed the monoculture control.The study's findings were as follows: In the nitrogen——phosphorus fertilizer addition test of annual grass and bean mixed sowing in alpine areas, nitrogen can increase the surface area and projected area of the root system, while phosphorus can increase the length of the root system and have significant improvement effects on soil nutrients. However, there is no clear benefit in terms of improving the forage's nutrient content. This could offer direction for choosing fertilizers and improving soil fertility in grasslands that are combined with annual grass and beans.

## Introduction

Nitrogen (N), as the most basic element for plant growth, had an important role in the process of plant growth and development, and was also an important limiting factor affecting plant

(2024-NK-P28) received by KD. This study was also financially supported by Qinghai Provincial Department of Science and Technology's Key R&D and Transformation Program in the form of a grant (2024-NK-137) received by KD.

**Competing interests:** The authors have declared that no competing interests exist.

productivity in terrestrial ecosystems [1]. The nitrogen addition could improve soil fertility and ensure a good crop harvest in the short term [2]. Oat was a nitrogen-loving crop, with the addition of nitrogen, the morphological characteristics such as root diameter and root length of oat would have significant changes, and the changes in total root length and root diameter affected the uptake of nitrate nitrogen [3]. Nitrogen addition could also affect the inter-root morphology of legumes [4], where the root surface area and root volume of legumes increase, however, the number of root forks decreases, suggesting that legumes inhibit the nitrogen-fixing ability of inter-root nitrogen-fixing bacteria in high-nitrogen-containing soils [5]. Phosphorus (P) is a non-renewable resource and one of the major nutrients necessary for plant growth, both as a constituent of organic compounds such as plant nucleic acids, Adenosine triphosphate (ATP), phospholipids and other organic compounds, as well as involved in metabolic processes such as energy transfer, protein activation and so on [6,7]. The level of phosphorus fertilizer application would directly affect plant metabolism, nutrient uptake, yield and nutrient formation [8]. However, irrational utilization of nitrogen and phosphorus could lead to the waste of resources, increase the loss of soil phosphorus, and the overuse of nitrogen was also a pollution threat to the natural environment [9,10].

The root system was an important organ connecting the plant to the soil and the main bridge for nutrient uptake and water transportation [11]. It helps plants to absorb nutrients from the soil through penetration and inter-root effects, and also transmits chemical signals from the soil to the plant through the root system [12]. The primary root of graminaceous plants was typically a shallow rooted fibrous root, which disappeared shortly after plant germination and was replaced by secondary roots of varying thickness, whose root system could extend in all directions of the soil, dividing the soil into fine grains to form a granular structure [13]. Legume root systems were more developed, belonging to the deep-rooted taproot type, mostly constituted of one main root and numerous lateral roots, with rhizomes on the roots that may fix nitrogen in the air in conjunction with rhizobacteria [14]. Shallow-rooted graminaceous plants can expedite soil nitrogen intake, hence increasing the nitrogen fixation capacity of deep-rooted legumes [15]. However, root interactions, nitrogen and phosphorus uptake, and root connections have not been well studied in field trials.

Based on the findings of a large number of previous experimental studies, the changes in yield, nutrients, and soil nutrient content of a mixture of grasses and legumes were investigated from 2020 to 2023, and it was discovered that the mixture had a significant increase in increasing forage yield and improving forage nutrients, as well as soil nutrient content, when compared to monoculture forage. However, the impact of exogenous fertilizer application on the interaction between pasture nutrients, root system, and soil nutrients remains unknown. Consequently, we planted 4,270 meters of mixed grassland with beans. In order to address the following two scientific concerns, we employed micro-root tube technology to measure the length, diameter, and surface area of the root systems of grass and legumes. We then merged these data with indexes relating to soil and pasture nutrients. When compared to feed with single plants, does mixed grassland enhance the biomass of plant roots? Is it possible for fertilizer with nitrogen and phosphorus to enhance and improve soil and forage nutrients in grass-bean mixes? (3) For varying rates of grass-bean combinations, which nitrogen and phosphorus fertilizer is the best option?

## Materials and methods

The experimental site was located at the Qinghai University Sanjiangyuan Ecosystem Ministry of Education Field Scientific Observatory, Chengdu Substation (33°24′30″N, 97°18′00″E), Chengduo County, Yushu Prefecture, Qinghai Province, at an altitude of 4,270 m. The area has a typical plateau continental climate with a long and cold winter.The average annual

temperature of the region was -5.6 °C ~ 3.8 °C, the extreme maximum temperature of about 25 °C, the extreme minimum temperature of about -30 °C, the four seasons are not clear, only 0 °C up and down into the hot and cold seasons, with no absolute frost-free period, the annual number of frosty days of about 260d, sunshine hours of 2,650.5h, the temperature will be gradually lowered with the increase in elevation, the annual average precipitation of 500 mm, precipitation is mainly distributed in June to September, accounting for about 75% of the annual precipitation. The average precipitation is 500 mm, and the precipitation is mainly distributed in June ~ September, accounting for about 75% of the annual precipitation. The soil of the test plot was alpine meadow soil, which was rich in humus, but the soil fertility was not high due to poor decomposition. Soil pH was 6.92, organic matter content was 2.36%, total nitrogen content was 9.50 g·kg⁻¹, quick-acting nitrogen content was 14.0 mg·kg⁻¹, total phosphorus content was 8.20 g·kg⁻¹, quick-acting phosphorus content was 7.0 mg·kg⁻¹, total potassium content was 13.50 g·kg⁻¹, quick-acting potassium content was 76.5 mg·kg⁻¹. The experimental site was not irrigated, and the previous crops were sown as a mixture of small rye and forage peas (Fig 1).

The oat variety was "Qing-tian No.1" and the forage pea variety was "Qing-jian No.1". Nitrogen fertiliser was urea (containing 46% N) at a dosage of 74.96 kg·hm⁻², and phosphorus fertiliser was calcium superphosphate (containing 12% P2O5) at a dosage of 299.85 kg·hm⁻². All the fertilisers were provided by the Grassland Institute of Qinghai Academy of Animal Husbandry and Veterinary Science.

The experiment commenced on 15 June 2023 and used a split-zone experimental design with the main zone being the fertiliser treatments recorded as $A_1$ (no fertiliser), $A_2$ (nitrogen fertiliser alone), $A_3$ (phosphorus fertiliser alone), and $A_4$ (Nitrogen-phosphorus mix). The subzones were mixed sowing proportions, recorded as $B_1$ (70:30), $B_2$ (50:50), and B3 (30:70), and single sowing control treatment groups were established, recorded as B4 (100:0) and B5 (0:100) (Table 1). Three replications were set up for each treatment and no artificial irrigation was applied throughout the reproductive period. The plot area was 15 m² (3m×5m). Before sowing, the test site was ploughed and harrowed, and sowing was carried out on the following day, the sowing method was peer strip sowing at a depth of 3–4 cm, and all plots were manually furrowed in 10 rows with a row spacing of 30 cm, and the fertiliser was spread by means of sowing, weighing the amount of 15 m² of fertiliser, and spreading it evenly into the test area. Harvesting was carried out on 25 September. The amount of graminaceous and leguminous sowing within each row was calculated by sowing in each single sowing treatment separately

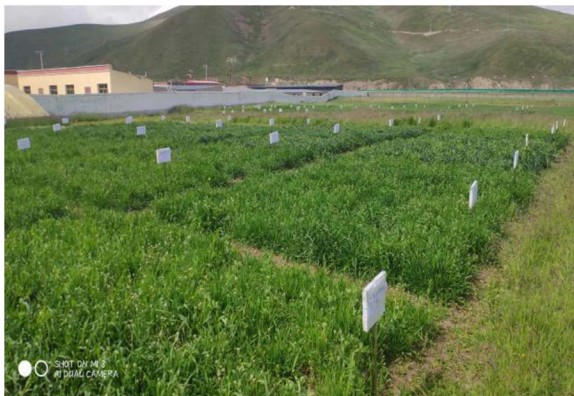

**Fig 1. Layout of oats and forage peas grown in this experimental site (elevation 4,270m, 15m² per treatment).**

**Table 1. Test plot and seeding amount of each treatment.**

| Treatment | | Seeding ratio (kg·hm$^{-2}$) | | Rate of fertilisation |
|---|---|---|---|---|
| | | *Avena sativa* | | *Pisum sativa L* |
| A$_1$ | B$_1$ | 157.54 | 22.60 | No fertiliser |
| | B$_2$ | 112.54 | 37.67 | |
| | B$_3$ | 67.54 | 52.74 | |
| | B$_4$ | 225.01 | – | |
| | B$_5$ | – | 75.34 | |
| A$_2$ | B$_1$ | 157.54 | 22.60 | CO(NH$_2$)$_2$,46%,N:74.96 kg·hm$^{-2}$ |
| | B$_2$ | 112.54 | 37.67 | |
| | B$_3$ | 67.54 | 52.74 | |
| | B$_4$ | 225.01 | – | |
| | B$_5$ | – | 75.34 | |
| A$_3$ | B$_1$ | 157.54 | 22.60 | P$_2$O$_5$ 12%,P:299.85 kg·hm$^{-2}$ |
| | B$_2$ | 112.54 | 37.67 | |
| | B$_3$ | 67.54 | 52.74 | |
| | B$_4$ | 225.01 | – | |
| | B$_5$ | – | 75.34 | |
| A$_4$ | B$_1$ | 157.54 | 22.60 | CO(NH$_2$)$_2$,46%,N:74.96 kg·hm$^{-2}$+P$_2$O$_5$ 12%,P:299.85 kg·hm$^{-2}$ |
| | B$_2$ | 112.54 | 37.67 | |
| | B$_3$ | 67.54 | 52.74 | |
| | B$_4$ | 225.01 | – | |
| | B$_5$ | – | 75.34 | |

The amount of seed required versus the amount of fertiliser applied under the four fertiliser treatments selected in this trial versus the five mixing ratio treatments.A$_1$=no fertiliser, A$_2$=nitrogen fertiliser alone, A$_3$=phosphorus fertiliser alone, and A$_4$=Nitrogen-phosphorus mix.The subzones were mixed sowing proportions, recorded as B$_1$=70:30, B$_2$=50:50, and B$_3$=30:70, and single sowing control treatment groups were established, recorded as B$_4$=100:0 and B$_5$=0:100.

[16]. For example, oats + forage pea mixed sowing graminaceous sowing = single sowing oats: 225.014 kg·hm$^{-2}$×70%=157.54 kg·hm$^{-2}$; forage pea sowing = single sowing forage peas: 75.34 kg·hm$^{-2}$×30%=22.60 kg·hm$^{-2}$ (Table 1).

During the sowing process on June 15, three polycarbonate root tubes were synchronized in each experimental plot. Three non-adjacent sowing rows were chosen for the arrangement, and the micro-root tubes, which have a diameter of 6.5 cm and a length of 100 cm, were used. In the experimental plot, a trench was dug during the arrangement, with a 45° downhill incline and a vertical depth of 55–60 cm. A layer of floating soil was then placed in the trench. In order to retain the soil near the exterior wall of the root tubes, the outer surface of the tubes was slightly sprayed with water after they were placed in the trench and covered with root tubes. To prevent light exposure that could stunt the root system's growth, the exposed portion of the root tube is wrapped in a black plastic bag and topped with a matching rubber cap. Once the pattern has been filled in, create a seed-sowing slit in the upper portion of the root canal, being cautious not to damage the root canal in the process.

Determination of forage nutrients: On 25 September 2023, three 1 m sample sections away from the side rows were randomly selected in each treatment and mowed flush with the ground, and the grass samples were placed in a drying oven for drying, and the plants were tested for Neutral Detergent Fibres (NDF); Acid Detergent Fibres (ADF) Crude Protein (CP) was determined by H2SO4- H2O2 digestion; Soluble Sugar (SS) was determined by anthrone

method; Total Nitrogen (TN) and Total Carbon (TC) were determined by elemental analyser. Total Phosphorus (TP), H2SO4-H2O2 decoction with molybdenum antimony antimony colourimetric method.

Plant root monitoring: from the 18th to the 20th of September on each piece of experimental land grass and legume forage grasses for the collection of root indicators, the use of CI-600 micro-root tubes on the root scanning and analysis, the first to open the top of the root tube of the plastic bag with a rubber cap, the use of matching cotton cloth will be the inside of the root tube of the water vapour to dry up, and wait for 5 minutes or so, will be ready to put a root scanner root scanner scanning the roots, the end of the end of the still good lid After finishing, the lid was still closed.The scanned images from the CI-600 were brought back to the laboratory, and the total root length (TRL), root surface area (RSA), root volume (RV), root average diameter (RAD), root projection area (RPA), and root projection area (RPA) were captured using WinRHZIO Tron MF (CID Bio-Science, Gamas, USA) software. Total Root Length (TRL), Root Surface Area (RSA), Root Volume (RV), Root Average Diameter (RAD), and Root Projection Area (RPA) were collected and recorded, and a total of 4,500 roots were traced by sampling three times in each experimental plot.

Soil nutrient content: Soil samples were collected from each experimental plot on 21 September, dried naturally, sieved for roots, stones and other debris, and then bagged for testing of soil total nitrogen (STN), soil quick-acting nitrogen (SQN), soil total phosphorus (STP), soil quick-acting phosphorus (SQP), soil organic carbon (SOC), and soil organic carbon (SOC). STP, Soil Quick-acting Phosphorus (SQP), Soil Organic Carbon (SOC). Among them, STN was determined by Kjeldahl digestion, SQN by alkaline diffusion, STP by NaOH melting-molybdenum-antimony antimony colorimetric method, SQP by sodium bicarbonate extraction-molybdenum-antimony antimony colorimetric method, and SOC by potassium dichromate-concentrated sulphuric acid external heating method.

## Computational and statistical analyses

Data were summarised using Microsoft Excel 2016 and IBM SPSS 23.0 was used to analyse the collected data and perform a two-factor ANOVA with general linear model (GLM) for main effects and two-factor interactions, and the differences between treatments were analysed using Duncan's multiple comparison test. Plotting was done using Origin 2021 software.

## Results and analyses

Fertilizer, Mixed sowing ratio, and Fertilizer × Mixed sowing ratio among pasture nutrients, plant roots, and soil nutrients all shown extremely significant differences (P<*0.01*), according to the two-factor ANOVA from Tables 2–4. Consequently, more multiple comparisons are required.

**Table 2. Two-factor ANOVA for forage quality by fertilisation and mix ratio.**

| Source of variation | F-value | | | | | | |
|---|---|---|---|---|---|---|---|
| | NDF | ADF | CP | SS | TN | TC | TP |
| Fertilise | 16.82** | 22.64** | 91.29** | 361.49** | 92.60** | 12.51** | 54.18** |
| Mixed sowing ratio | 88.67** | 73.91** | 272.34** | 331.16** | 185.90** | 6.89** | 35.47** |
| Fertiliser × Mixed sowing ratio | 25.40** | 44.72** | 60.05** | 238.90** | 91.35** | 25.01** | 61.46** |

Note:

*indicates significant difference (P<*0.05*),

**indicates highly significant difference (P<*0.01*).NDF=Neutral Detergent Fibres, ADF=Acid Detergent Fibres, CP=Crude Protein, SS=Soluble Sugar, TN=Total Nitrogen, TC=Total Carbon, TP=Total Phosphorus.

**Table 3. Two-factor analysis of variance (ANOVA) of fertiliser and mix ratio on plant root system.**

| Source of variation | F-value | | | | |
|---|---|---|---|---|---|
| | TRL | RSA | RV | RAD | RPA |
| Fertilise | 111.38** | 186.10** | 5.48** | 1.87 | 32.70** |
| Mixed sowing ratio | 76.30** | 46.92** | 85.73** | 810.45** | 419.11** |
| Fertiliser × Mixed sowing ratio | 47.76** | 9.66** | 20.00** | 9.51** | 5.63** |

Note:

*indicates significant difference (P<0.05),

**indicates highly significant difference (P<0.01).TRL=Total Root Length, RSA=Root Surface Area, RV=Root Volume, RAD=Root Average Diameter, RPA=Root Projection Area.

**Table 4. Two-factor analysis of variance (ANOVA) of soil nutrients by fertiliser and sowing mix ratio.**

| Source of variation | F-value | | | | |
|---|---|---|---|---|---|
| | STN | SQN | STP | SQP | SOC |
| Fertilise | 58.19** | 165.63** | 8.99** | 1041.76** | 75.19** |
| Mixed sowing ratio | 41.75** | 111.55** | 47.45** | 102.78** | 71.39** |
| Fertiliser × Mixed sowing ratio | 127.72** | 208.52** | 138.91** | 220.84** | 146.37** |

Note:

*indicates significant difference (P<0.05),

**indicates highly significant difference (P<0.01).STN=Soil Total Nitrogen,SQN=Soil Quick Nitrogen,STP=Soil Total Phosphorus,SQP=(Soil Quick-acting Phosphorus,SOC=Soil Organic Carbon.

The variations in feed nutrients under various fertilization treatments were shown results (Fig 2). One-way ANOVA revealed that, in comparison to A2 and A4, the CP and TN contents were increased by 47.75% and 67.75% (P<0.05), respectively, while the NDF and ADF contents of A1 were relatively high. A2, A4 had comparatively higher SS and TC contents than A1, A3. A2's SS content increased by 35.13%, 19.75%, (P<0.05), while A4's increased by 41.40%, 25.31%, (P<0.05) compared to A1, A3. Compared to A3, TC content A2 rose by 1.90% (P<0.05). Comparing A1 to A2, A3, A4, 22.08%, 34.64%, and (P<0.05), the TP content of A1 increased by 21.59%.The findings indicated that while there were no discernible differences in the forage nutrients under different fertilization treatments in the grass-bean mixed sowing grassland, the overall presentation of forage nutrients among the non-fertilized treatments was improved.

Fertilizer addition enhanced the plant root system length greatly (Fig 3). Compared to A1, A2, and A4, the TRL value of the A3 treatment increased by 13.26%, 5.20%, 5.61%, (P<0.05), and the TRL value of the A2 and A4 treatments increased by 7.66%, 7.24%, (P<0.05).In comparison to A1, A3, and A4, the RSA value of the A2 therapy increased by 62.33% (P<0.05) and by 9.64% (32.51%) (P<0.05).Comparing A2 to A1, the RPA value increased by 23.43% (P<0.05). The aforementioned univariate analysis results showed that fertilization treatments could boost root biomass, with the A2 treatment offering the most benefits over the other fertilization treatments.

Exogenous fertilizers had a significant impact on soil nutrients (Fig 4). STN content increased by 19.42% and 14.96% (P<0.05) in the A3 treatment compared to A2 and A4, respectively, and SQN increased by 14.24% to 21.62% (P<0.05) in the A3 treatment compared to the other fertilization treatments. When comparing the three fertilization treatments to A1, the SQP content rose by 153.37% to 181.39% (P<0.05). the SOC content of the A3 therapy rose by 15.51% to 19.37%. The aforementioned data' analysis indicates that applying fertilizer

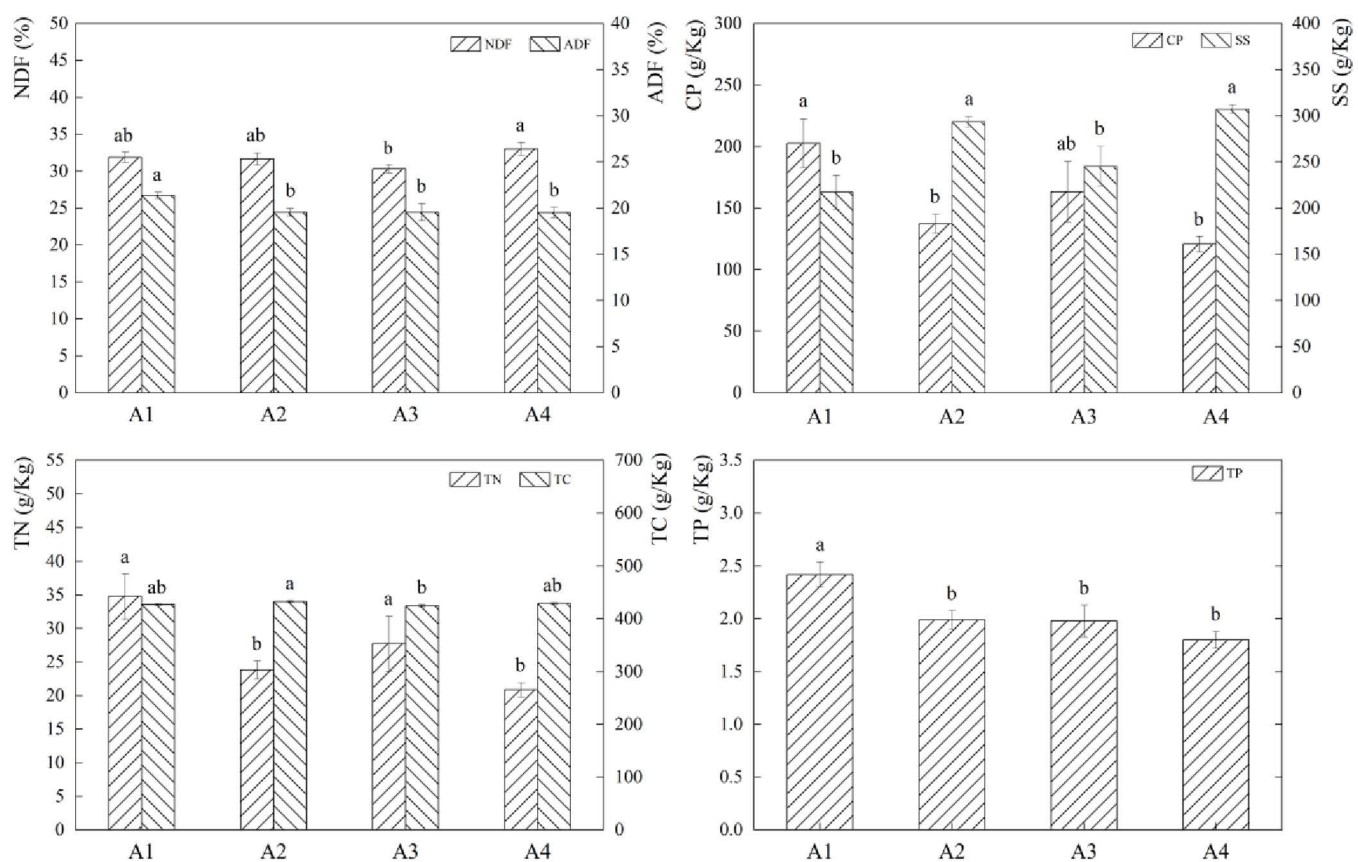

**Fig 2. Effects of different fertilisation treatments on forage nutrients.** The bar graph indicates the data distribution of each index, different symbols of a and b indicate significant differences among forage nutrient indexes under different fertilisation conditions, while the same symbols indicate no significant differences.

can raise the amount of each nutrient in the soil, with the A3 treatment exhibiting a higher rise in nutrient content.

With varying mix proportions, the percentage of legume forage increased while the forage NDF and ADF contents dropped (Fig 5). In the B1 treatment, the NDF contents rose by 13.21% to 23.57% ($P<0.05$), while in the B2 treatment, the ADF contents climbed by 13.44% to 23.83% ($P<0.05$). A progressive increase in the content of CP and SS occurred in tandem with the rise in the percentage of legume forage. When compared to the combination treatments, the B4 and B5 treatments had the highest TN and TC content. Increases in TN content B4 treatment were 58.54% to 87.10% ($P<0.05$) and 75.05% to 106.58% ($P<0.05$) in TC content B4 treatment.

Forage root features were improved by mixed treatments (Fig 6), with a greater dominance of forage root characteristics at increasing percentages of graminoid forage. The B1 treatment increased TRL by 5.53% to 10.93% ($P<0.05$), RSA by 20.60% to 33.90% ($P<0.05$), RV by 54.04% to 94.82% ($P<0.05$), and RAD by 113.81% to 120.64% ($P<0.05$). RPA increased from 103.78% to 136.21% ($P<0.05$) in comparison to the B4 and B5 treatments. When graminoids made up a greater percentage of the grasses in the mixed sward, the forage root system was significantly improved when compared to the monoculture treatment, according to the combined results of the aforementioned tests.

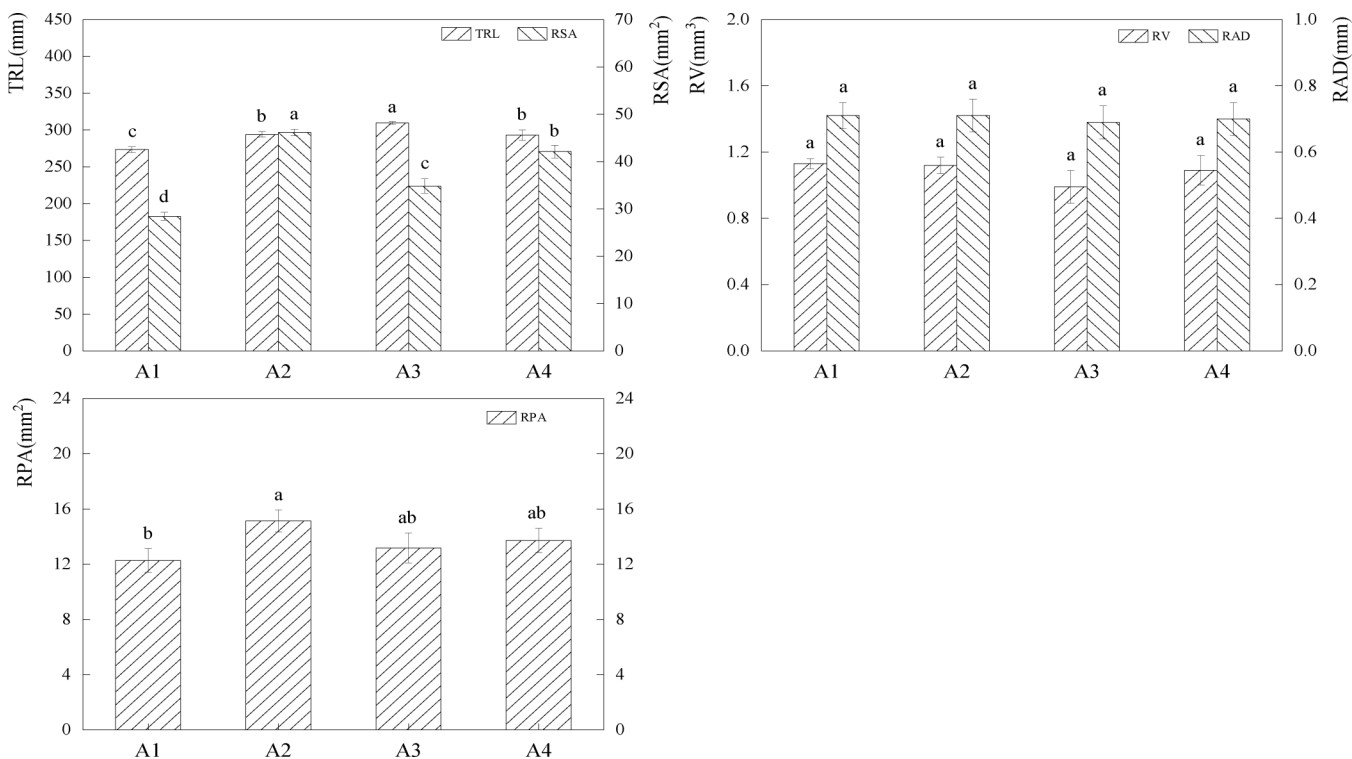

**Fig 3. Effects of different fertilisation treatments on forage root system.** The bar graph indicates the data distribution of each index, different symbols of a and b indicate significant differences among forage root system indexes under different fertilisation conditions, while the same symbols indicate no significant differences.

the STN content rose 12.12% to 17.85% in the B4 treatment and 11.29% to 16.96% in the B5 treatment when compared to the mix (Fig 7); SQN content rose 8.88% to 18.96% in the B4 treatment and 12.92% to 23.37% in the B5 treatment; STP content rose 5.76% to 9.87% in the B4 treatment and 5.49% to 9.60% in the B5 treatment; SQP content increased 5.33% to 16.40% in the B4 treatment; and SOC content increased 18.56% to 22.66% (P<*0.05*). Additionally, when the percentage of legume forage grew, so did the overall trend of forage soil nutrients.

When fertilizer application and mixing proportion were combined, there was no discernible change in the nutrients in the fodder (Fig 8). The NDF and ADF contents would gradually decline as the percentage of legume forage rose. The highest NDF content was found in the A4B2 treatment, which was higher than that of the monocropped oat treatment (from 10.85% to 28.52%, P<*0.05*) and higher than that of the monocropped forage pea (from 22.16% to 50.46%, P<*0.05*).With the rise in the percentage of legumes, the contents of CP, SS, TN, TC, and TP all exhibited an overall increasing trend. This suggests that the phase interaction treatment of fertilizer application and mix fraction had no effect on pasture nutrients.

During the interaction, the mixed treatments had greater forage root indices than the monoculture treatments (Fig 9), and the root system shrank as the amount of legume increased.RSA increased by 22.52% to 90.80%, 9.06% to 130.78%, RV by 70.47% to 311.98%, 23.40% to 698.98%, RAD by 124.88% to 136.18%, 76.70% to 190.12%, and RPA by 63.98% to 140.29%, 107.88% to 222.35%, and TRL by 5.25% to 30.03% and 5.59% to 20.92% as a result of A3B2 therapies. According to the aforementioned findings, fertilization treatment could also

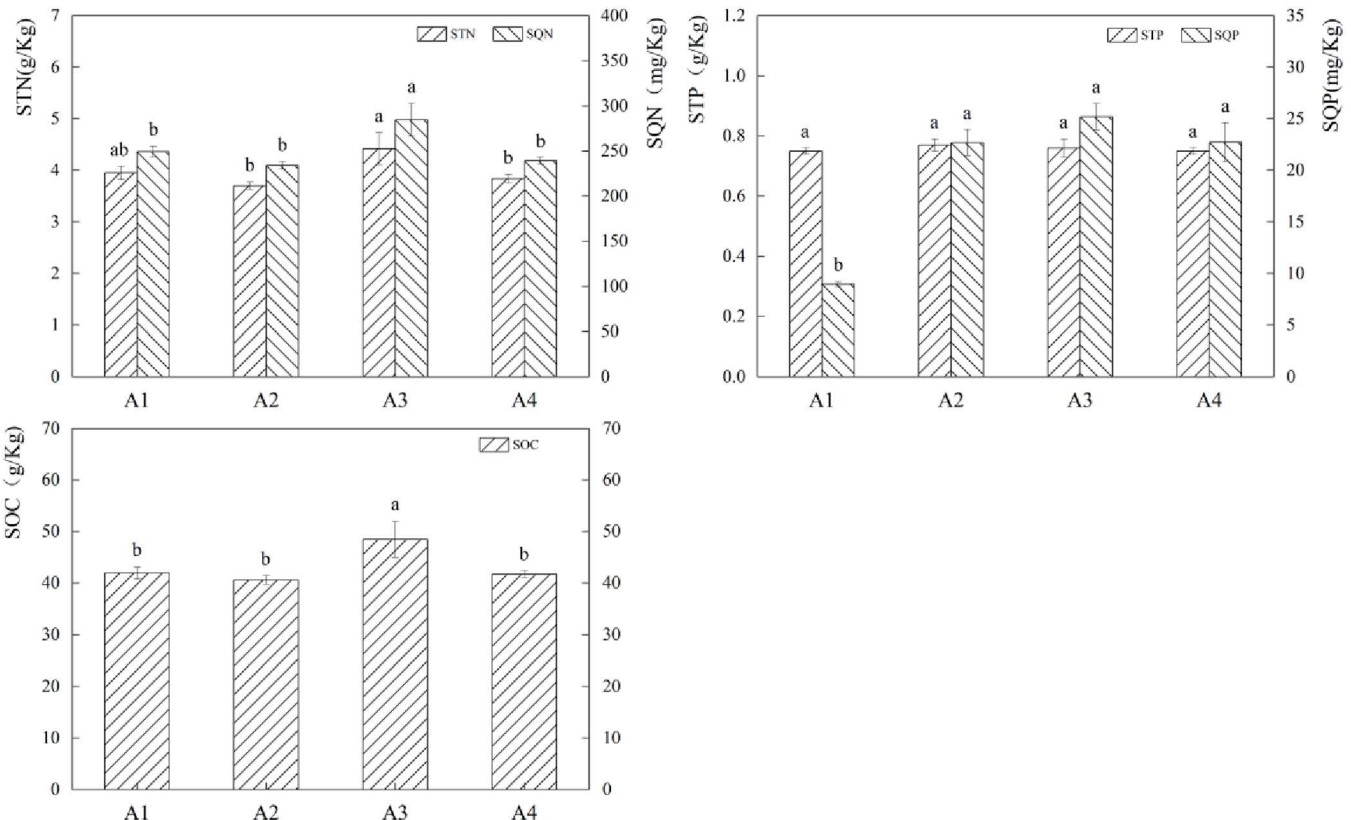

**Fig 4. Effects of different fertilisation treatments on soil nutrients.** The bar graph indicates the data distribution of each index, different symbols of a and b indicate significant differences among forage soil nutrient indexes under different fertilisation conditions, while the same symbols indicate no significant differences.

enhance the plant's root morphology traits, and mixed sowing was more effective than single sowing in increasing the plant's root morphology.

The highest levels of STN, SQN, STP, SQP, and SOC were found in A3B4 and A3B5 treatments under the interaction treatments. In comparison to other unicorn oat and forage pea treatments, STN contents increased by 59.68% to 114.30% and 8.44% to 83.98%, SQN by 4.27% to 92.90%, and 62.58% to 77.49%; STP increased by 13.07% to 50.30%, 17.10% to 42.07%; SQP increased by 9.01% to 270.60%, 37.79% to 290.40%; and SOC increased by 61.95% to 121.69%, 16.81% to 95.41%. It suggests that adding phosphorus fertilizer by itself will raise the soil's concentration of each nutrient (Fig 10).

Forage ADF, NDF, and CP did not significantly correlate with the root system and soil nutrients, according to correlation analysis (Fig 11). On the other hand, there was a negative correlation between the changes in SS content and soil nutrients and a positive correlation with the root system. Conversely, TN, TP, and TC had a positive correlation with soil nutrients and a negative correlation with roots. While there was a positive link between the indicators in the root system and the soil nutrients, there was a negative correlation between the root system indicators overall and the soil nutrients. This suggests that the root system encouraged root growth by absorbing nutrients from the soil. The entire soil system was affected by decreased nutrient content, which could additionally have an impact on plant TN, TP, and TC levels.

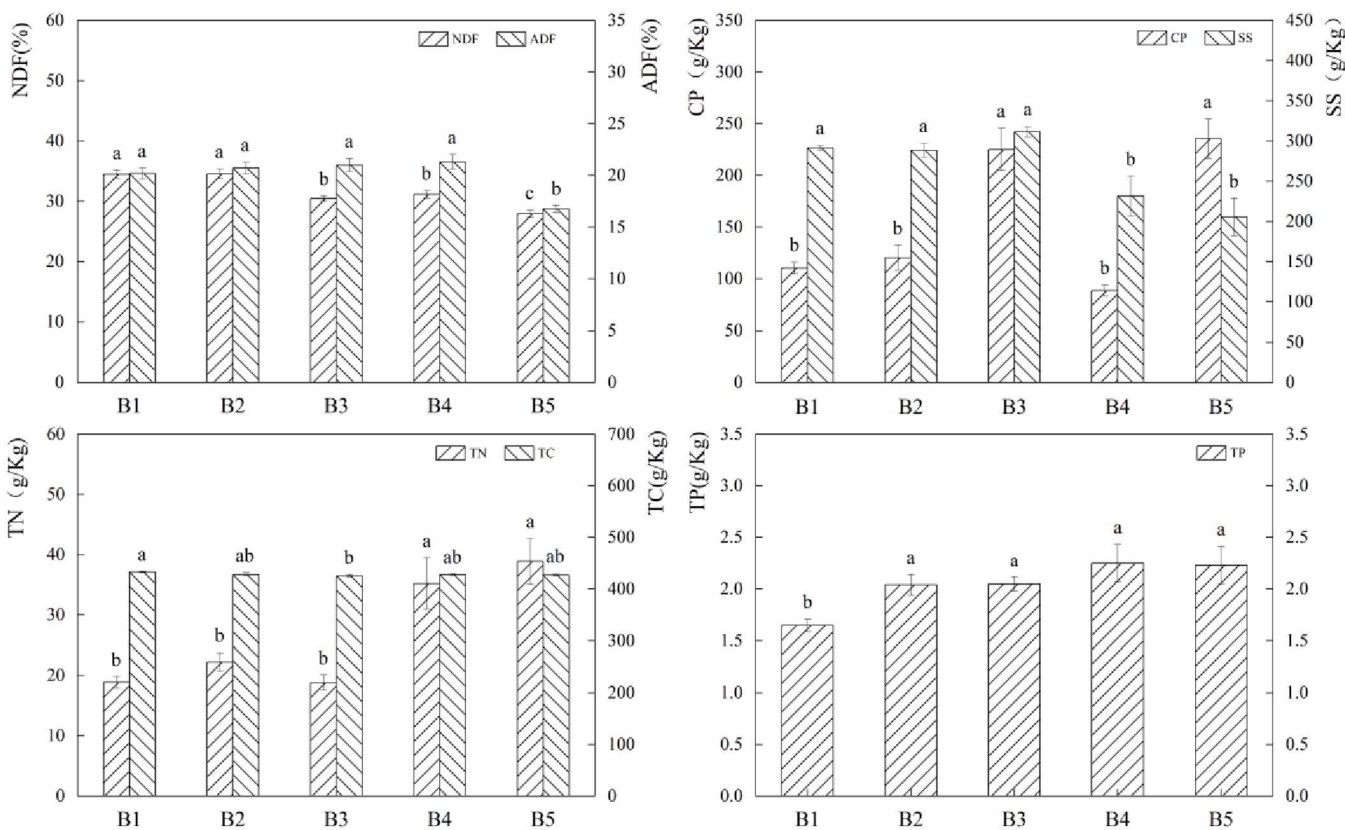

**Fig 5. Effects of different mixing ratios on forage nutrients, the bar graph indicates the distribution of data of each index, the different symbols of a and b indicate that there is a significant difference between forage nutrient indexes in different mixing ratios, and the same symbols indicate that there is no significant difference.**

## Disscussion

The quality of forage nutrients not only affects the healthy growth and development of livestock, but also has a significant impact on processed livestock products.Some studies have shown that fertiliser application affects grass quality, and the appropriate amount of fertiliser can increase crude protein content [17]. However, this paper found that there was no significant difference in crude protein content between different fertiliser treatments through the results of fertiliser application in different treatments at an altitude of 4270 m. Moreover, some treatments had a better advantage over mixed treatments between monocultures, which may be due to the following reasons: firstly, the altitude of the experimental site affected the development of forage grasses, which resulted in a shorter fertility period. Secondly, the effect of climatic conditions at the trial site, where the trial started in June and ended in October, when the forage was no longer undergoing growth, may have resulted in low nutrient accumulation in the plants.Thirdly, the nutrient content of the soil at the experimental site resulted in low soil microbial activity in alpine regions due to shallow soil layers and low temperatures [18], and the fact that nutrient loss in alpine soils> soil residues> plant uptake [19] may also explain the fact that there was no significant difference between no fertiliser and fertiliser treatments in this paper.

Fertilisation can significantly increase root biomass in grassland ecosystems [20], and nitrogen and phosphorus affect the root system of plants by influencing root length and

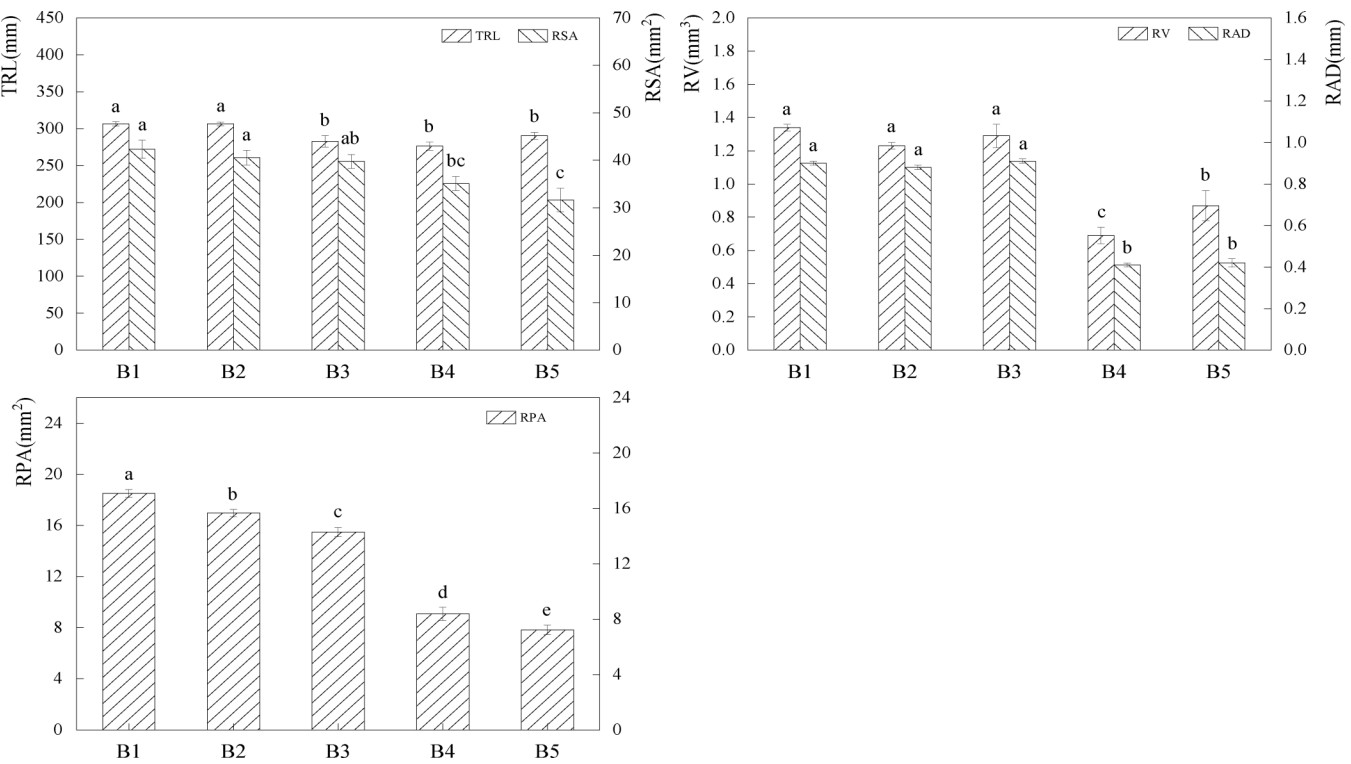

**Fig 6. Effects of different mixing ratios on forage root system, the bar graph indicates the distribution of data of each index, different symbols of a and b indicate significant differences between forage root system indexes in different mixing ratios, while the same symbols indicate no significant differences.**

biomass [21]. In this experiment, there were some differences in the effects on the root system between nitrogen and phosphorus fertilisers, nitrogen fertiliser increased the root surface area (RSA) and root projected area (RPA), while phosphorus fertiliser increased the length of the root system (TRL). The nitrogen——phosphorus mixture treatments did not show more obvious advantages, and the root morphology of all three fertiliser treatments was better than that of no fertiliser. Nitrogen——phosphorus mixing increases the mortality of roots in the surface soil and decreases the mortality of roots in the deeper layers [22]. Because the experiment of this paper was located in the alpine region, the soil layer is shallow, the water loss in the shallow soil is serious, and most of the root system is in the shallow soil, which may explain the conclusions of this paper, or it may be due to the change of the surface moisture layer, under the condition of low moisture, the internal result and function of the root system produces a significant change, and these changes lead to accelerated death of roots, which affects the number of root mortality in the surface soil layer [23].

Nutrient content of soil plays a vital role in plant growth and metabolism [24], and this study showed that phosphorus addition increased soil total nitrogen (STN) and quick-acting nitrogen (SQN) content. Fertiliser treatments also increased soil total nitrogen (STP) and quick-acting phosphorus (SQP) content, and fertiliser application leads to an increase in the accumulation of phosphorus in the soil [25], and as phosphorus is more readily available in the soil for decomposition and absorption, this also contributes to an increase in the total phosphorus content of the soil. The addition of nitrogen and phosphorus nutrients promotes the reaction between soil phosphorus and microorganisms, which enables microorganisms

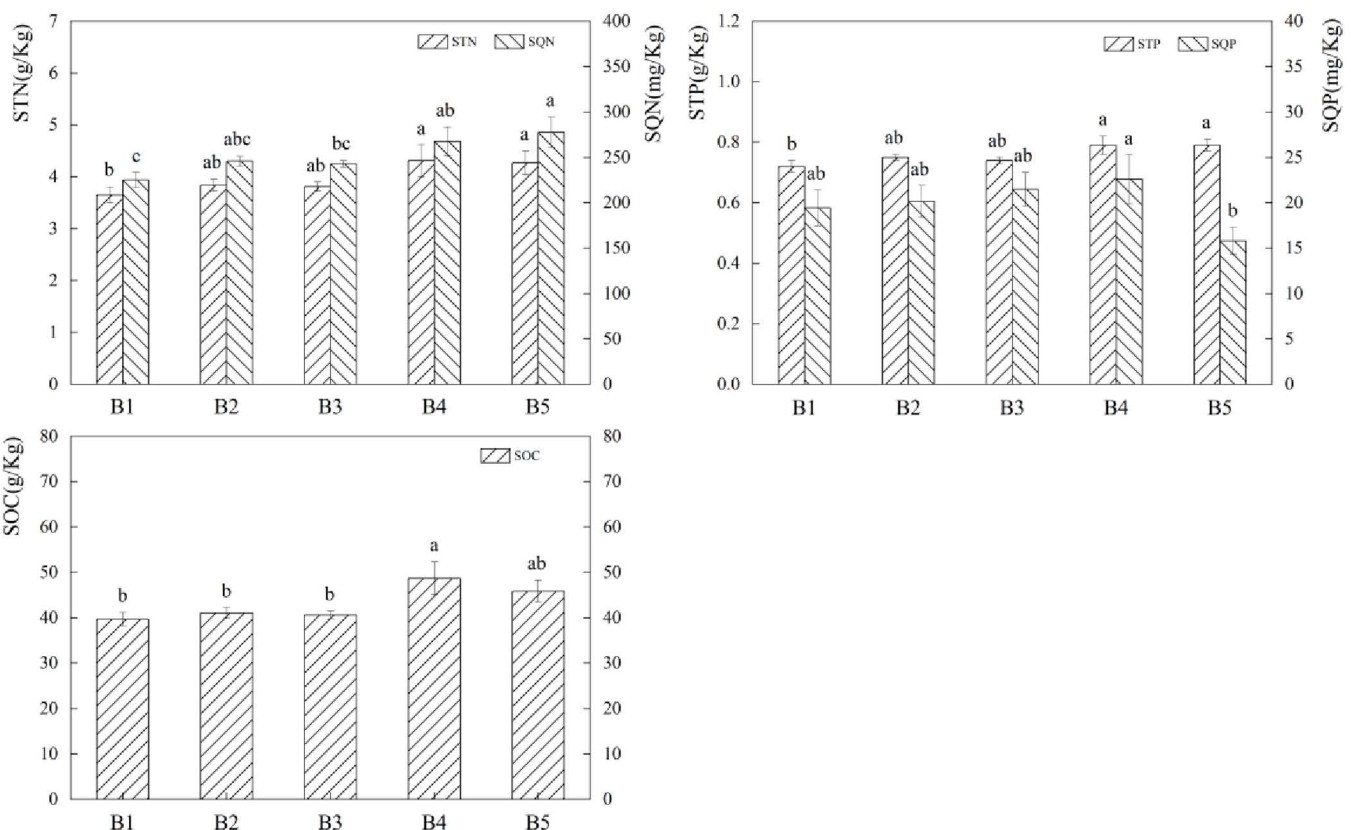

**Fig 7. Effects of different mixing ratios on soil nutrients, the bar graph indicates the distribution of data of each index, different symbols of a and b indicate significant differences between soil nutrient indexes of forage in different mixing ratios, while the same symbols indicate no significant differences.**

to sequester more phosphorus [26], and also contributes to the increase of phosphorus in the soil. Most of the effects of exogenous nutrient addition on soil properties are positively correlated effects [27], so the author believes that it may also be related to the climate of the alpine region, where lower temperatures reduce the rate of mineralisation, and when exogenous nutrients are added, they may increase microbial activity, which leads to an increase in each of the soil nutrients.

Gramineous-bean mixed sowing artificial grassland quality and increase yield was an important way, compared with single sowing grassland, mixed sowing treatment of forage nutrients will have different degrees of improvement because it can make full use of space, water, light and other resources. However, it was also influenced by a variety of factors such as species, geographic location, and mixing ratio. In this study, forage NDF and ADF contents showed a decreasing trend with the increase of legume forage proportion, while CP and SS contents showed a gradual increasing trend. NDF and ADF contents affected the dry matter intake and digestibility of livestock [28–29], and there was a negative correlation between them, which also indicated that the increase of legume forage content improved the intake and digestibility of forage. The main role of legume forage was to increase the CP content of forage, and because of the nitrogen fixation effect of its own roots could also promote the overall development of the mixed sown grassland, but too large a proportion would affect the forage yield, indicating that an increase in the proportion of legume forage instead led to a decrease in forage yield [30]. TN, TC, and

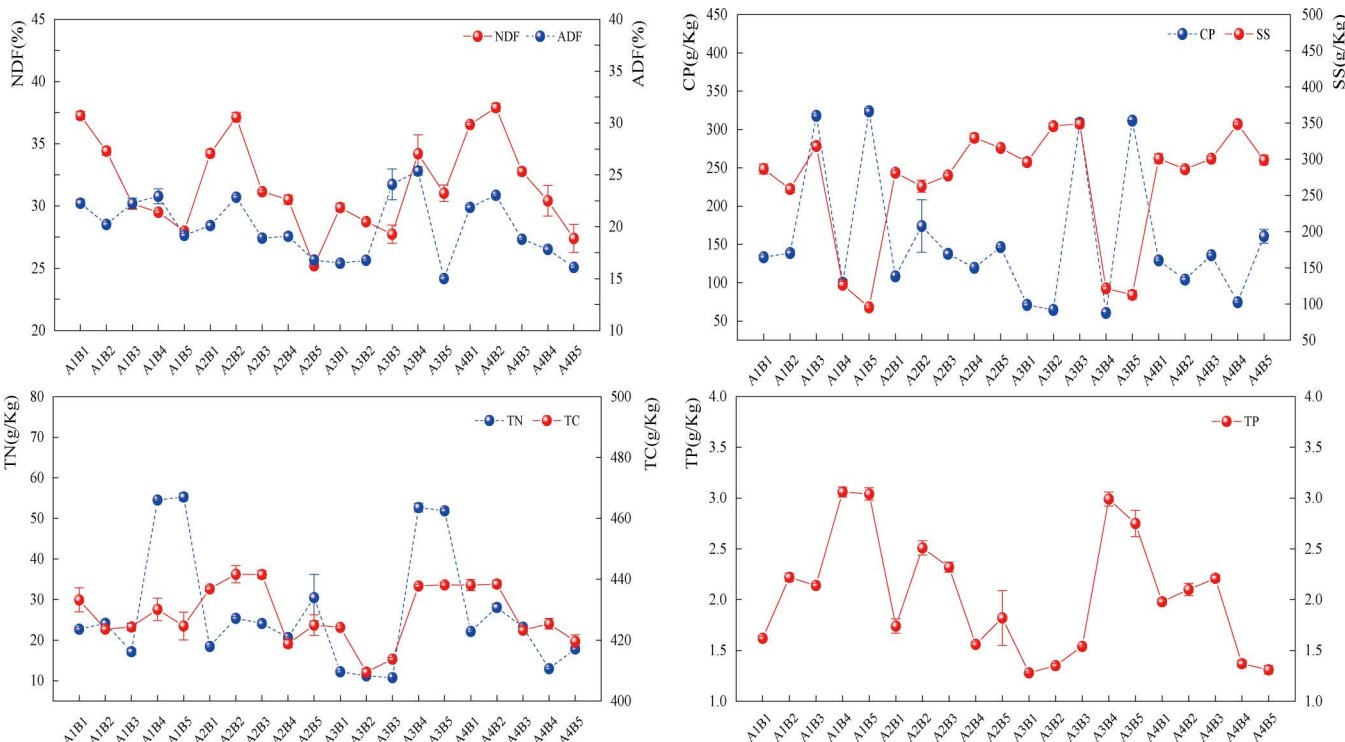

**Fig 8. Effect of fertiliser and ratio interaction on forage nutrients,** The line graphs represent the distribution of data for each indicator of forage neutral detergent fibre (NDF), acid detergent fibre (ADF), crude protein (CP), soluble sugars (SS), plant total nitrogen (TN), plant total carbon (TC), plant total phosphorus (TP), and the different dots represent the mean values and trends between forage nutrient indicators in the fertiliser × proportion interaction.

TP in forage all showed higher in the monoculture treatment, which might be related to the climate of the experimental site, where the recycling of nutrients due to lower temperatures resulted in lower nitrogen and phosphorus content in the plants [31].The author suggests that this may explain the higher plant growth vigour in the mixed sown grassland than in the single sown treatment, as more energy was required to maintain higher growth vigour as the temperature gradually decreased, suggesting that the rate of nutrient conversion in the plants of the mixed sown grassland was higher than that of the single sown treatment.

Root system was an important organ for plants to acquire soil nutrients [32], and the results of this study showed that the root morphology was better when the proportion of graminaceous family was higher, and it decreased gradually with the decrease of the proportion. Root length and root surface area reflect the spatial expansion of the root system and the rate of nutrient and water uptake [33], which was similar to the results of the present study, and thus also indicates that the ability to acquire nutrients is higher when the proportion of graminaceous species is higher, and the adaptability to the severe ecological environment of the alpine region is also higher.Larger root volume, mean diameter and projected area also indicate a wider root system in the soil with greater metabolic capacity and extension [34], which also further suggests that grass-bean mixed grassland under 70:30 treatment has better survivability and better nutrient uptake in the soil, and comparing to monoculture treatment can also indicate better resilience of the mixed treatment.

The results of this study showed that the nutrient contents in soil under different mixing ratios showed a gradual increase with the increase in the proportion of legume, indicating

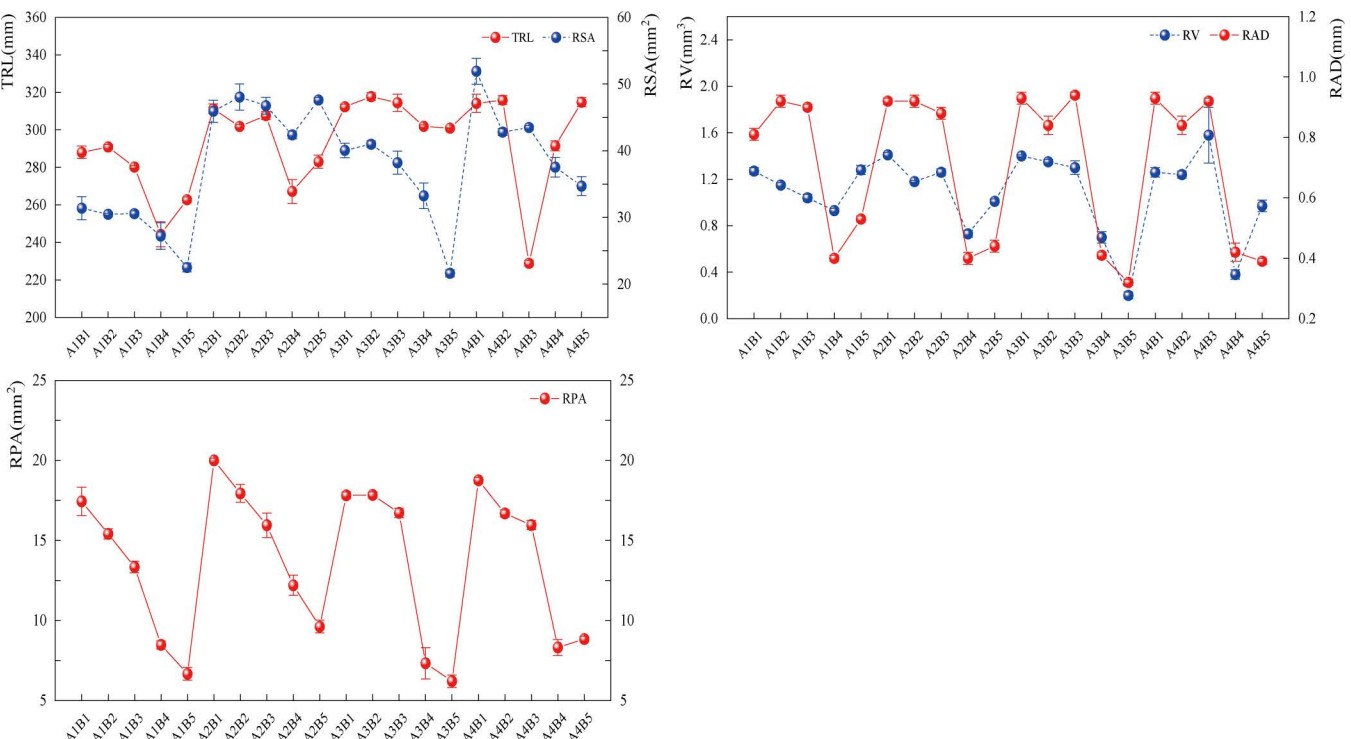

**Fig 9. Effect of fertiliser and ratio interaction on forage root system, Line graphs represent the distribution of data for each indicator of total root length (TRL), root surface area (RSA), root volume (RV), root average diameter (RAD), and root projected area (RPA) of forage, and the different points represent the mean values and trends of forage root indicators in the fertiliser × ratio interaction.**

that the mixing treatment could absorb the nutrients in soil more effectively, and the 70:30 combination of grass and legume forage had the lowest nutrient contents in soil, with significant differences between the control and the monoculture treatment, which also indicated that the plant activity might be relatively higher under the treatment, and the rate of nutrient absorption would be faster. This also suggests that the plant activity may be relatively higher and the rate of nutrient uptake faster under this treatment. However, there was no significant difference with the other mixed treatments, which also indicated that this treatment promoted better nitrogen fixation by the legume forage and higher microbial activity in the soil.

Through the interaction treatment of fertiliser application and mix proportion, the results of forage nutrients in this paper showed that there was no significant difference between forage nutrients among treatments, and the NDF and ADF contents showed a gradual decreasing trend as the proportion of leguminous forage increased, which was different from the conclusions of our experiments in the previous years.In different combinations and proportions interaction, the content of NDF and ADF of uni-sown treatment was lower than that of mixed sowing 5.13% to 32.54 and 9.26% to 33.42% [35], which may be closely related to whether or not fertiliser was applied to the experimental site, and we found that fertiliser application did not have a significant effect on the nutrients of forage in mixed treatments in alpine areas, although it improved the nutrients of uni-sown treatments, which also indicated that fertiliser application treatments increased the nutrients of the forage,. However, the nitrogen fixation of legume forage in the grass-bean mix can provide the required nutrients for the grass, which can reduce the

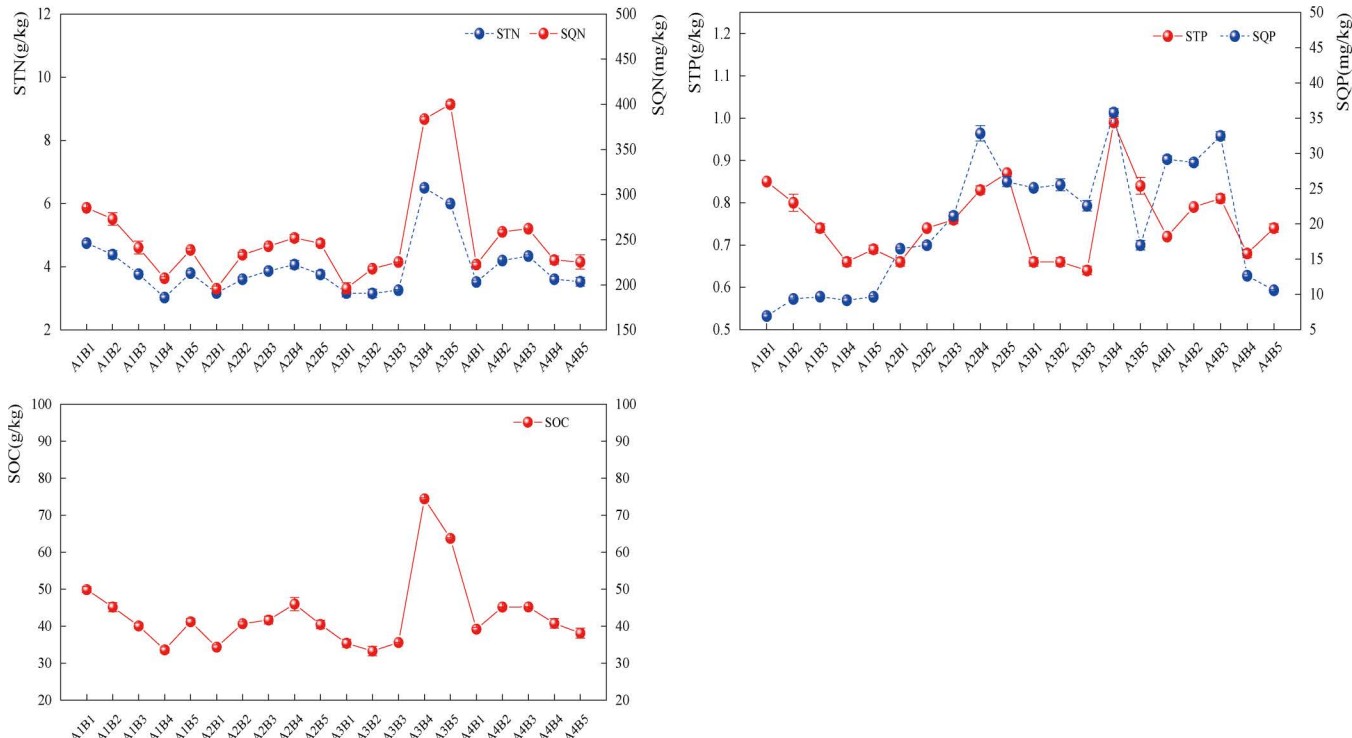

**Fig 10. Effect of fertiliser and ratio interaction on soil nutrients, The line graphs represent the distribution of data of soil total nitrogen (STN), soil quick-acting nitrogen (SQN), soil total phosphorus (STP), soil quick-acting phosphorus (SQP), and soil organic carbon (SOC), and the different dots represent the mean and trend of forage soil nutrient indexes in the fertiliser ×ratio interaction.**

fertiliser dosage appropriately in the grass mix. The content of CP, SS, TN, TC and TP of forage showed a gradual increase in the proportion of legumes, indicating that legumes have obvious advantages in improving the nutrient content of forage in the mixed sowing treatment.

In this study, we found that the TRL and RSA of the root systems of the three fertilisation treatments increased significantly under the interaction effect, but the TRL, RSA, RV and RPA of the root systems gradually decreased with the increase of the proportion of legume, indicating that too large a proportion of legume forage will affect the plant root system, and the pressure of the inter-species competition will gradually increase, which may be mainly due to the increase of legume, nitrogen fixation effect, etc., will promote the growth of the root system of oats. However, the shallow soil layer in the experimental area led to an increase in the number of oats, which would limit the space for the growth of the legume root system and ultimately lead to a reduction in root biomass. Gramineae inhibit the growth of legume roots by increasing root length and volume to encroach on legume root space and compete for legume nutrients.

The analysis of the results of soil nutrient content of each nutrient showed that with the increase of legume percentage of the nutrient content under the non-fertilised treatment were gradually reduced, and the mixed treatment was higher than the uni-sown treatment, which also showed that the soil nutrient content under the mixed treatment was better than the uni-sown treatment, and also showed that the mixed sowing could improve the soil. Nitrogen demand combined with the nitrogen fixation of legumes can promote the soil-ecosystem cycle, single application of phosphorus fertiliser monoculture treatment was significantly

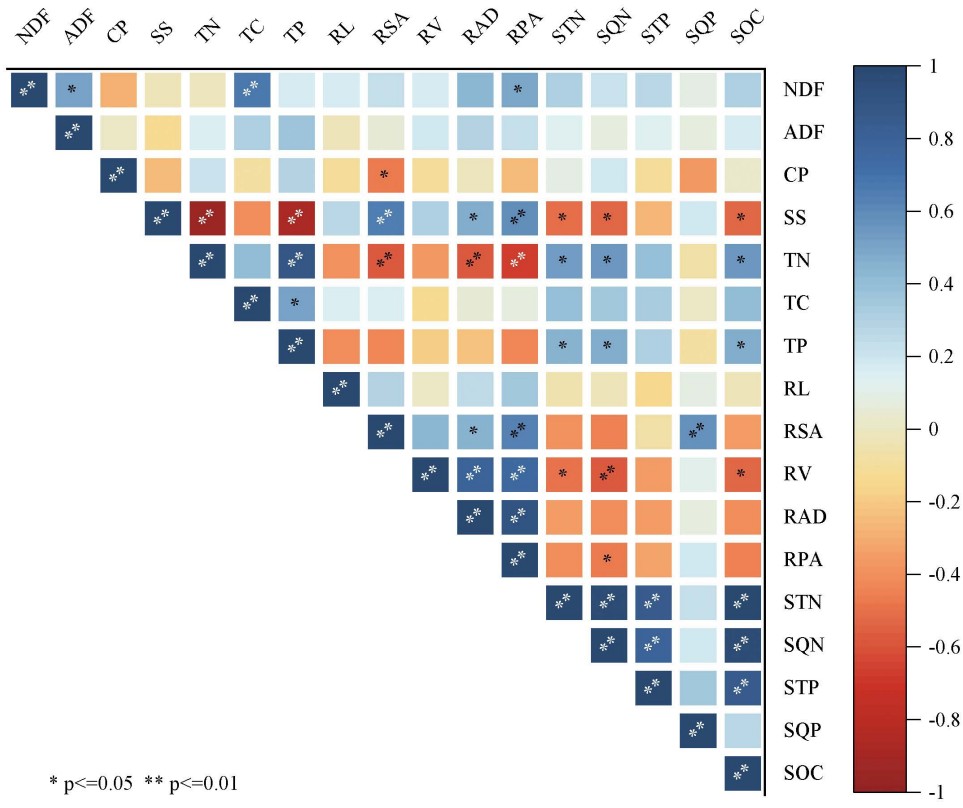

**Fig 11. Correlation analysis between forage nutrients/root characteristics/soil nutrients, * indicates significant difference (P<0.05),** indicates highly significant difference (P<0.01), blue indicates positive correlation, red indicates negative correlation.**

higher than the other treatments, monoculture treatment in the soil residual STN, SQN, STP, SQP, SOC nutrients had higher content, which also indicates that the monoculture treatment did not completely absorb the nutrients, which may contain most of the unabsorbed as well as the return of the.The main reason may be derived from the climatic factors of the test area, due to the cold climate of the area, the relative activity of soil microorganisms may be low under the monoculture treatment, resulting in a large portion of under-utilisation. In contrast, the relatively higher microbial vigour under the mixed sowing treatment, together with the synergistic promotion between grasses and legumes, would maintain the nutrient content in the soil in a relatively balanced state, which would also contribute to the maintenance of ecological balance.

## Conclude

The study's outcomes may offer useful information for alpine annual grass-bean mixed grasslands that want to add phosphorus and nitrogen. Both phosphorus and nitrogen fertilizers have little effect on the nutrients in forage, suggesting that phosphorus fertilizers will have a bigger impact on the soil and roots of the forage while nitrogen fertilizers have no effect on the nitrogen fixation effect of legumes.In contrast to monoculture grasslands, legume forage enhances soil fertility and forage nutrients in grass-bean mixes. It also increases plant root biomass. Legumes should not, however, make up an excessive amount of the diet; a 70%:30%

ratio for grass-bean combinations was suggested. In alpine settings, it could be used as an experimental foundation for comparing production performance as well as for widely popularizing and using grass-bean combinations of various types.

## Supporting information

**S1 Data. Data analysis.**
(ZIP)

## Author contributions

**Data curation:** Wei-shan Lin, Lin Zhang.

**Investigation:** Fei Li, Xi-jie Wei.

**Project administration:** KEJIA DE.

**Supervision:** Xue-mei Xiang.

**Writing – original draft:** Ting-xu Feng.

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
