## [Decision Letter · Decision Letter 0]

22 Jan 2025

PONE-D-24-51070Influence of nitrogen and phosphorus additions on nutrient uptake in mixed annual grass and bean sown grassland in alpine regionPLOS ONE

Dear Dr. DE,

Thank you for submitting your manuscript to PLOS ONE. After careful consideration, we feel that it has merit but does not fully meet PLOS ONE’s publication criteria as it currently stands. Therefore, we invite you to submit a revised version of the manuscript that addresses the points raised during the review process.

We look forward to receiving your revised manuscript.

Kind regards,

Paulo H. Pagliari

Academic Editor

PLOS ONE

Journal Requirements:

Additional Editor Comments:

Please use the reviewer comments to prepare a revised manuscript

Reviewers' comments:

Reviewer's Responses to Questions

**Comments to the Author**

1. Is the manuscript technically sound, and do the data support the conclusions?

Reviewer #1: Yes

2. Has the statistical analysis been performed appropriately and rigorously?

Reviewer #1: Yes

3. Have the authors made all data underlying the findings in their manuscript fully available?

Reviewer #1: Yes

4. Is the manuscript presented in an intelligible fashion and written in standard English?

Reviewer #1: Yes

5. Review Comments to the Author

Reviewer #1: General Comments

This paper investigates the effects of nitrogen and phosphorus fertilizer addition on the nutrition, root characteristics, and soil nutrients in artificial mixed grasslands (grasses and legumes) in alpine regions. The results show that the addition of nitrogen and phosphorus fertilizers does not significantly enhance the nutritional value of forages in mixed grasslands, and no fertilization appears to be more beneficial for the growth of grass-legume mixes. Nitrogen fertilization increases root surface area, while phosphorus fertilization improves root length. Additionally, the mixed planting significantly enhances root characteristics and soil nutrients. The optimal sowing ratio is 70% grass and 30% legumes, which balances the enhancement of forage nutrition and soil nutrient sustainability, providing important insights for grassland management and sustainable development in alpine regions. This study is worthy of being published. However, some revisions that the authors can be considered may be needed.

Main Comments

Although the overall aim of the paper is to investigate the multiple impacts of nitrogen and phosphorus fertilization on mixed grasslands, the research question is presented in a broad manner and lacks specificity. For example, does the study primarily focus on improving forage nutrition, changes in soil nutrients, or optimizing root characteristics?

Minor Comments

1. When discussing the mechanisms of nitrogen fertilization in increasing soil nutrients (e.g., total nitrogen and available nitrogen), the paper could incorporate the following reference to explore the contribution of microbial activity to nutrient accumulation: doi: 10.1016/j.apsoil.2024.105498.

2. In the background and discussion of soil nutrient responses, the study could be linked with this meta-analysis to analyze the general patterns of nitrogen addition on alpine grasslands and highlight how regional differences or fertilization quantities may have contributed to the specific results:Fu Gang, Shen Zhenxi. DOI: 10.1016/j.apsoil.2017.02.016.

3. When discussing plant responses to nitrogen fertilization in relation to the special climate conditions of the study area, the paper could refer to the following study to support general conclusions about plant responses in alpine regions: doi:10.1007/s00344-016-9595-0.

4. The sentence “The addition of nitrogen and phosphorus nutrients did not significantly advantage the enhancement of forage nutrients in grass-bean mixes compared to no fertilization, without fertilization could be more beneficial to the growth of grass-bean mixes” is overly complex and unclear. A clearer version might be: "The addition of nitrogen and phosphorus did not significantly enhance forage nutrients in grass-legume mixes compared to no fertilization, and no fertilization appeared more beneficial for the growth of grass-legume mixes."

5. In the phrase "nitrogen-phosphorus mixing," a long dash should be used for clarity and formality: "nitrogen—phosphorus mixing."

6. The citation format for references should be consistent. Some references include URLs while others do not.

7. "The effect of phosphorus was studied by applying phosphorus fertilizer to the plots" could be rephrased as: "The effect of phosphorus was studied through the application of phosphorus fertilizer."

8. "Based on the experiment, it is concluded that..." can be simplified to: "The experiment concluded that..."

9. The figure captions should not only label the figures but also attempt to explain the overall content and findings of the images.

6. PLOS authors have the option to publish the peer review history of their article (what does this mean?). If published, this will include your full peer review and any attached files.

Reviewer #1: No

---

## [Author Response · Author response to Decision Letter 1]

23 Feb 2025

Dear editor,

Thanks for your letter and for reviewers' comments concerning our manuscript entitled"Influence of nitrogen and phosphorus additions on nutrient uptake in mixed annual grass and bean sown grassland in alpine region "(Manuscript ID: PONE-D-24-51070). Those comments are allvaluable and helpful for revising and improving our paper. We havestudied all comments carefully and have made conscientious correction. The revised parts have been marked in the paper.The responses to the reviewers' comments are as follows:

Comment 1.Please ensure that your manuscript meets PLOS ONE's style requirements, including those for file naming. The PLOS ONE style templates can be found at.

Response:We really appreciate your efforts and comments on our manuscript. We have reviewed the formatting issues in the manuscript in light of the suggestions and have made changes with reference to the URL you provided.

Comment 2.We note that the grant information you provided in the ‘Funding Information’ and ‘Financial Disclosure’ sections do not match.

Response:We noticed that you mentioned a mismatch in some of the grant information, so we've reworked it. Confirm that the grant information we provided is correct.

Comment 3.We note that your Data Availability Statement is currently as follows: [All relevant data are within the manuscript and its Supporting Information files

Response:Based on your suggestions, we have compiled the data in the article and will send it to you by email.

Comment 4.Although the overall aim of the paper is to investigate the multiple impacts of nitrogen and phosphorus fertilization on mixed grasslands, the research question is presented in a broad manner and lacks specificity. For example, does the study primarily focus on improving forage nutrition, changes in soil nutrients, or optimizing root characteristics?

Response:We note that you have raised the issue of the broad and unfocussed formulation of your research question. After careful consideration, we have revised the title of the article to ‘Nitrogen and phosphorus fertilisers optimise root morphology and soil nutrients in mixed annual grass and bean sown grassland in alpine regions’ to refine the research question. sown grassland in alpine regions’ to refine the research question.

Comment 5.When discussing the mechanisms of nitrogen fertilization in increasing soil nutrients (e.g., total nitrogen and available nitrogen), the paper could incorporate the following reference to explore the contribution of microbial activity to nutrient accumulation: doi: 10.1016/j.apsoil.2024.105498.

Response:We really appreciate your efforts and comments on our manuscript. Thinking that the comments you gave had a very important positive effect on the article, we listened to you and added the literature you recommended.

Comment 6. In the background and discussion of soil nutrient responses, the study could be linked with this meta-analysis to analyze the general patterns of nitrogen addition on alpine grasslands and highlight how regional differences or fertilization quantities may have contributed to the specific results:Fu Gang, Shen Zhenxi. DOI: 10.1016/j.apsoil.2017.02.016.

Response:We really appreciate your efforts and comments on our manuscript. Thinking that the comments you gave had a very important positive effect on the article, we listened to you and added the literature you recommended.

Comment 7.When discussing plant responses to nitrogen fertilization in relation to the special climate conditions of the study area, the paper could refer to the following study to support general conclusions about plant responses in alpine regions: doi:10.1007/s00344-016-9595-0.

Response:We really appreciate your efforts and comments on our manuscript. Thinking that the comments you gave had a very important positive effect on the article, we listened to you and added the literature you recommended.

Comment 8.The sentence “The addition of nitrogen and phosphorus nutrients did not significantly advantage the enhancement of forage nutrients in grass-bean mixes compared to no fertilization, without fertilization could be more beneficial to the growth of grass-bean mixes” is overly complex and unclear. A clearer version might be: "The addition of nitrogen and phosphorus did not significantly enhance forage nutrients in grass-legume mixes compared to no fertilization, and no fertilization appeared more beneficial for the growth of grass-legume mixes."

Response:We really appreciate your efforts and comments on our manuscript. Based on your suggestions, we have made the appropriate changes in the article.

Comment 9.‘In the phrase "nitrogen-phosphorus mixing," a long dash should be used for clarity and formality: "nitrogen—phosphorus mixing."

Response:We really appreciate your efforts and comments on our manuscript. Based on your suggestions, we have made the appropriate changes in the article.

Comment 10.The citation format for references should be consistent. Some references include URLs while others do not.

Response:Thank you very much for your efforts and comments on our manuscript. We have rechecked the references and reworked them according to the editorial formatting requirements.

Comment 11"The effect of phosphorus was studied by applying phosphorus fertilizer to the plots" could be rephrased as: "The effect of phosphorus was studied through the application of phosphorus fertilizer."

Response:We really appreciate your efforts and comments on our manuscript. Based on your suggestions, we have made the appropriate changes in the article.

Comment 12 "Based on the experiment, it is concluded that..." can be simplified to: "The experiment concluded that..."

Response:We really appreciate your efforts and comments on our manuscript. Based on your suggestions, we have made the appropriate changes in the article.

Comment 13 The figure captions should not only label the figures but also attempt to explain the overall content and findings of the images.

Response:We really appreciate your efforts and comments on our manuscript. Based on your suggestions, we have made the appropriate changes in the images of the article.

---

## [Decision Letter · Decision Letter 1]

5 Mar 2025

Nitrogen and phosphorus fertilisers optimise root morphology and soil nutrients in mixed annual grass and bean sown grassland in alpine regions

PONE-D-24-51070R1

Dear Dr. DE,

We’re pleased to inform you that your manuscript has been judged scientifically suitable for publication and will be formally accepted for publication once it meets all outstanding technical requirements.

Kind regards,

Paulo H. Pagliari

Academic Editor

PLOS ONE

Additional Editor Comments (optional):

We can now accept your manuscript for publication. Congratulations!

Reviewers' comments:

Reviewer's Responses to Questions

**Comments to the Author**

1. If the authors have adequately addressed your comments raised in a previous round of review and you feel that this manuscript is now acceptable for publication, you may indicate that here to bypass the “Comments to the Author” section, enter your conflict of interest statement in the “Confidential to Editor” section, and submit your "Accept" recommendation.

Reviewer #1: (No Response)

2. Is the manuscript technically sound, and do the data support the conclusions?

Reviewer #1: (No Response)

3. Has the statistical analysis been performed appropriately and rigorously?

Reviewer #1: (No Response)

4. Have the authors made all data underlying the findings in their manuscript fully available?

Reviewer #1: (No Response)

5. Is the manuscript presented in an intelligible fashion and written in standard English?

Reviewer #1: (No Response)

6. Review Comments to the Author

Reviewer #1: The author has made a great deal of revisions according to the comments, and I think it is acceptable now.

7. PLOS authors have the option to publish the peer review history of their article (what does this mean?). If published, this will include your full peer review and any attached files.

Reviewer #1: No

---

## [Editor Report · Acceptance letter]

PONE-D-24-51070R1

PLOS ONE

Dear Dr. DE,

I'm pleased to inform you that your manuscript has been deemed suitable for publication in PLOS ONE. Congratulations! Your manuscript is now being handed over to our production team.

Kind regards,

on behalf of

Dr. Paulo H. Pagliari

Academic Editor

PLOS ONE
